# Interactions of Glutamatergic Neurotransmission and Brain-Derived Neurotrophic Factor in the Regulation of Behaviors after Nicotine Administration

**DOI:** 10.3390/ijms20122943

**Published:** 2019-06-16

**Authors:** Jieun Kim, Ju Hwan Yang, In Soo Ryu, Sumin Sohn, Sunghyun Kim, Eun Sang Choe

**Affiliations:** 1Department of Biological Sciences, Pusan National University, 63-2 Busandaehak-ro, Geumjeong-gu, Busan 46241, Korea; jieun0479@pusan.ac.kr (J.K.); juhwanyang@pusan.ac.kr (J.H.Y.); insoo.ryu@kitox.re.kr (I.S.R.); soomin@pusan.ac.kr (S.S.); s.hyun@pusan.ac.kr (S.K.); 2Research Center for Safety Pharmacology, Korea Institute of Toxicology, 141 Gajeong-ro, Yuseong-gu, Daejeon 34114, Korea

**Keywords:** glutamate receptor, neurotrophic factor, nicotine dependence, TrkB, striatum

## Abstract

Nicotine causes tobacco dependence, which may result in fatal respiratory diseases. The striatum is a key structure of forebrain basal nuclei associated with nicotine dependence. In the striatum, glutamate release is increased when α7 nicotinic acetylcholine receptors expressed in the glutamatergic terminals are exposed to nicotine, and over-stimulates glutamate receptors in gamma amino-butyric acid (GABA)ergic neurons. These receptor over-stimulations in turn potentiate GABAergic outputs to forebrain basal nuclei and contribute to the increase in psychomotor behaviors associated with nicotine dependence. In parallel with glutamate increases, nicotine exposure elevates brain-derived neurotrophic factor (BDNF) release through anterograde and retrograde targeting of the synapses of glutamatergic terminals and GABAergic neurons. This article reviews nicotine-exposure induced elevations of glutamatergic neurotransmission, the bidirectional targeting of BDNF in the striatum, and the potential regulatory role played by BDNF in behavioral responses to nicotine exposure.

## 1. Introduction

Nicotine is a major constituent of the tobacco plant and commercial cigarettes, and is considered to be the cause of compulsive tobacco smoking [1,2]. Nicotine delivered to the striatum increases extracellular dopamine and glutamate concentrations by stimulating nicotinic acetylcholine receptors (nAChRs) [3,4,5,6]. An increase in the level of extracellular dopamine in the striatum in turn increases glutamate release by stimulating basal ganglia [7,8,9]. Resulting elevated glutamate levels stimulate ionotropic- and metabotropic glutamate receptors (iGluRs and mGluRs) expressed in striatal gamma amino-butyric acid (GABA)ergic neurons [10,11,12]. These stimulations cause short- and long-term changes in neuronal activities and gene expressions and cause behavioral changes by upregulating GABAergic outputs to other parts of basal ganglia [7,13,14].

Nicotine exposure upregulates expression of brain-derived neurotrophic factor (BDNF) in glutamatergic terminals and GABAergic neurons of the striatum [15,16,17]. Furthermore, evidence demonstrates that BDNF gene expression is upregulated by increased glutamate release in the striatum [16,18,19,20]. Cleaved BDNF is the most abundant neurotrophic factor in the brain and binds to mature-BDNF-activated tropomyosin receptor kinase B (TrkB) and pro-BDNF-stimulated p75 receptors [21,22,23]. This study was undertaken to review spatiotemporal interactions of glutamate and BDNF in the striatum that modulate psychomotor and nicotine-seeking behaviors in response to nicotine exposure.

## 2. The Striatum Potentiates Glutamate Release after Nicotine Administration

In the human brain, the striatum is a key structure of forebrain basal ganglia and is composed of dorsal and ventral striatum. The dorsal striatum is comprised of the caudate nucleus and putamen (CPu), and the ventral striatum of the nucleus accumbens (NAc) [8,24]. The CPu receives dopaminergic inputs from the substantia nigra pars compacta in the midbrain and glutamatergic inputs mainly from the somatosensory cortex [24,25]. On the other hand, the NAc receives dopaminergic inputs from the ventral tegmental area (VTA) in the midbrain and glutamatergic inputs from the prefrontal cortex (PFC) [25]. GABAergic medium spiny neurons constitute >90% of the striatum, integrate glutamatergic and dopaminergic inputs, and potentiate glutamatergic response when a drug of abuse is administered [7,24,25].

Previous studies have shown chronic administration of nicotine increases extracellular dopamine concentrations in rat striatum [26]. Other studies have shown that dopamine release is increased by the local infusion of nicotine into the CPu and NAc as determined by in vitro fast-scan cyclic voltammetry analysis [27]. Repeated nicotine exposure increases extracellular glutamate concentrations in rat CPu and NAc as determined by real-time glutamate biosensing and microdialysis [13,27]. Chronic administration of nicotine increases dopamine release in the striatum [27,28]. Systemic repeated exposure to nicotine dose-dependently increases dopamine levels in the CPu and NAc [29], and that local infusion of nicotine increases glutamate release in the striatum [6,30]. These findings suggest that nicotine administration potentiates glutamate release in the GABAergic neurons of the striatum by integrating dopaminergic neurotransmission.

## 3. Stimulation of α7 nAChRs in the Striatum is Required for Glutamate Release

N-AchRs are a heterogeneous family of pentameric ion channels ubiquitously expressed in the central nervous system [31,32,33]. They are composed of eight α (α2-7, 9, 10) and three β (β2–4) subunits which include heteromeric (α4)_3_(β2)_2_, (α4)_2_(β2)_3_, (α3)_2_(β4)_3_, α4α6β3(β2)_2_, and homomeric (α7)_5_ [31,32,33]. Striatal exposure to nicotine by cigarette smoking binds and stimulates α4β2 and/or α7 nAChRs and increases glutamate release [32,34]. Numerous studies have demonstrated nicotine administration potentiates glutamate release by stimulating excitatory α7 nAChRs dominantly expressed in glutamatergic terminals of the striatum [6,35,36,37]. For instance, blockade of α7 nAChRs decreases increases in glutamate release in the CPu induced by repeated nicotine exposure [13]. Whereas blockade of α4β2 nAChRs reduced the nicotine-induced increase in dopamine release in the CPu and NAc [27,29]. These findings suggest α7 nAChRs stimulation plays a crucial role in enhancing striatal glutamate release, and this suggestion is supported by finding that blockade of α7 or α4β2 nAChRs in the CPu by local infusion of the receptor antagonist, methyllycaconitine (MLA) or dihydro-β-erythroidine DhβE, respectively, reduced the nicotine-induced increase in glutamate release [6]. It has also been reported that α7 nAChRs blockade reduced the nicotine-induced increase in glutamate release in the CPu innervating neurons from the PFC [5,38,39]. However, the stimulation of α4β2 nAChRs in the CPu by indirect glutamate release caused by stimulating dopaminergic systems via basal ganglia in the forebrain has not been elucidated. Together, these findings suggest α7 stimulation is responsible for potentiating glutamate release in the CPu and upregulating GABAergic outputs to basal ganglia.

Repeated systemic injections of nicotine were found to increase behavioral sensitization, as determined by assessing locomotor and stereotypy activities [13]. Furthermore, a challenge injection following repeated exposure to nicotine also showed increased behavioral sensitization [13]. This increase in behavioral sensitization was reduced by pharmacological blockade of α7 nAChRs in the CPu of rats [13]. Similarly, anxiety-like behaviors caused by nicotine withdrawal were attenuated by α7 nAChR blockade in mice [40]. In addition, motivation induced by nicotine self-administration in rats was reduced by α7 nAChR blockade induced by intra-NAc infusion of the receptor antagonist, PNU282987 [41]. Taken together, these findings suggest nicotine-induced α7 nAChR stimulation in the striatum is required to sensitize psychomotor behaviors, such as locomotor and stereotype activities.

## 4. Stimulation of Glutamate Receptors in GABAergic Output Neurons Increases Behavioral Sensitization and Nicotine-Seeking Behavior

Elevation of glutamate release due to α7 nAChR stimulation over-stimulates glutamate receptors in GABAergic neurons of the striatum [42]. Two families of glutamate receptors expressed in the striatum are iGluRs and mGluRs [43,44,45]. Stimulation of the iGluRs, *N*-methyl-d-aspartate receptor (NMDAR), α-amino-3-hydroxy-5-methyl-4-isoxazolepropionic acid receptor (AMPAR), and kainate receptor (KAR) in the striatum is known to upregulate Ca^2+^ and Na^+^ conductance [43,44,45,46]. Previous studies have demonstrated that increased glutamate release in the striatum alters the functions of iGluRs [26,30]. Repeated nicotine administration increases the phosphorylation of the NMDAR GluNR2B subunit in the NAc [47,48]. Furthermore, electrophysiological recordings obtained from rat NAc reveals that local nicotine infusion increases NMDA currents in neurons [42,49]. These findings suggest that stimulation of iGluRs by elevated glutamate release after nicotine administration leads to changes in neuronal excitability and subsequent behavioral changes. This suggestion is supported by findings that increases in nicotine-seeking behavior in response to nicotine self-administration are associated with phosphorylation of the NMDAR GluNR2B and AMPAR GluR2 subunits in the NAc [47,48]. While blockade of NMDAR in the VTA suppresses the nicotine-induced increase in self-administration [50].

The mGluRs are classified into group I, II, and III in accord with the pharmacology and biochemistry of GTP-binding proteins [44]. Stimulation of mGluR5 linked to excitatory Gq, which is dominantly expressed in the striatum, upregulates the productions of inositol trisphosphate (IP_3_) and diacylglycerol (DAG) via the hydrolysis of phospholipase C (PLC)-mediated phosphatidylinositol 4,5-bisphosphate (PIP_2_) [44]. In turn, IP_3_ increases Ca^2+^ release from the ER, while DAG activates protein kinase C (PKC) and upregulates Ca^2+^ signaling cascades, and thus, induces behavioral changes [9]. Growing evidence shows that blockade of mGluR5 in the striatum reduces nicotine-induced increases in locomotor activity, motivation, conditioned place preference, and self-administration [51,52,53,54]. These findings support that stimulation of mGluR5 in the striatum by glutamate increase is required for nicotine-induced behavioral sensitization and nicotine-seeking behavior.

Group II (mGluR2/3) and III (mGluR4/6-8) expressed mainly in the necks of glutamatergic terminals are also stimulated by excessive glutamate release [42,55,56]. These receptors linked to inhibitory G_i/o_ control glutamate release by inhibiting adenylyl cyclase-mediated signaling cascades [42,55,56]. Previous studies have demonstrated stimulation of group II/III mGluRs attenuates nicotine-induced self-administration by reducing glutamate release in the CPu and NAc [57]. Pharmacological stimulation of these receptors or their potentiation by peptides, such as mGluR2 PAMs AZ8418, and AZD8529, in the medial forebrain reduces nicotine-induced increases in self-administration [58]. These findings suggest stimulation of group II/III mGluRs downregulates nicotine-induced increase in glutamatergic neurotransmission by reducing glutamate release in a feedback control manner.

## 5. Nicotine Increases Bidirectional BDNF Release in the Striatum

Nicotine increases BDNF expression in the striatum [16,19,20], but the mechanisms responsible have not been elucidated. BDNF is constitutively transcribed and synthesized in the ER as a prepro-(35 kDa), and pro-peptide (32 kDa) [59]. Rodent BDNF genes contain nine exons (I-IX) which include their own promoters [60]. The promoter region of exon IV contains three Ca^2+^-responsive element (CaRE1-3 also called CRE1-3), and interaction between CaRE3 and phosphorylated CREB increases the transcriptional activity of BDNF on promoter at exon IV of the BDNF gene and induces the translation of pro-BDNF [61].

Pro-BDNF is then proteolytically processed through the trans-Golgi network (TGN) by endopeptidases, such as the Golgi-resident- and secretory granule-resident proprotein convertases 1–7 (PC), or the membrane-anchored protein, furin [61,62]. Cleaved BDNF in vesicles associated with either PC or furin undergoes posttranslational modifications, such as *N*-acetylation and carboxyl-terminal amination [63], which are transported to glutamate terminals. During this process, phosphorylated Huntingtin (htt), which is activated by a variety of TGN signaling cascades, recruits the motor proteins, kinesin 1 and dynein [64,65,66]. Binding of the htt complex to vesicles determines whether the targeting of cleaved BDNF is anterograde or retrograde [64,65,66]. Cleaved BDNF is known to be sorted by two different types of vesicles, that is, secretion vesicles and secretory granules, which are linked to furin and PC, respectively [61,62]. In addition, secretory granules contain TGN-originated sorting protein complex in which cleaved BDNF is released in a pH-dependent manner. Subsequent exocytosis triggered by the stimulation of α7 nAChRs by nicotine binding then results in the release of a mixture of pro- and mature-BDNF [23]. Released pro-BDNF is further regulated by endopeptidases, such as tissue-plasminogen activator (tPA) plasmin cascade or matrix-metalloproteinases (MMPs), which are located outside of neurons [67,68]. These extracellular enzymes convert pro-BDNF into mature-BDNF, which stimulates TrkB in striatal GABAergic neurons. However, not all pro-BDNF is converted to mature-BDNF [59].

Apart from anterograde BDNF release, cleaved BDNF is also constitutively transported to dendritic spines of GABAergic neurons via retrograde axonal transport. The stimulation of glutamate receptors by nicotine-induced glutamate release facilitates BDNF secretion into synaptic clefts [69]. Although the cellular mechanisms responsible for the retrograde release of BDNF from the dendritic spines in the striatum are not clear, they may involve increased Ca^2+^ influx due to iGluR stimulation [70,71]. Stimulation of mGluR5 may contribute to BDNF release by activating PLC-mediated IP_3_, which increases Ca^2+^ mobilization from the ER, and DAG, which activates PKC and increases Ca^2+^ conductance by stimulating NMDAR [72]. Furthermore, an increase in intracellular Ca^2+^ concentration may activate the SNAP REceptor (SNARE) proteins, synpatotagamin-6 (SYT6) and complexin, and facilitate the retrograde targeting of BDNF release [73].

## 6. Nicotine Activates BDNF-Mediated TrkB Signaling Cascades in GABAergic Neurons

BDNF released by either anterograde or retrograde targeting binds to its receptor tropomyosin receptor kinase B (TrkB), also known as type 2 of neurotrophic tyrosine kinase [23], whereas pro-BDNF binds to p75 neurotrophin receptor (p75NTR), though it may be converted to mature-BDNF by MMPs [67]. Previous studies have shown nicotine increases BDNF levels and activates TrkB-mediated signaling cascades in GABAergic neurons [74]. Furthermore, BDNF infusion into the NAc stimulates TrkB phosphorylation [16], that is, binding of BDNF to TrkB results in the phosphorylations of tyrosine and serine of TrkB at positions Tyr515, Tyr816, and Ser478 [75]. Phosphorylation at Tyr515 is linked to Shc adaptor protein and activates the Pi3K/Akt pathway [76,77], and Shc activation upregulates Ras-signaling cascades [75]. The activation of Pi3K/Akt and Ras-signaling pathways leads to ERK phosphorylation [76,77], which in turn results in the phosphorylation of CREB and the activation of mammalian target of rapamycin (mTOR)-signaling cascades [75,76,77]. In addition, phosphorylation at Tyr816 creates a binding site for PLCγ, and TrkB-PLCγ binding results in increases of Ca^2+^ release from the ER by activating IP_3_ [75]. This release of Ca^2+^ into the cytoplasm then activates a variety of Ca^2+^ signaling cascades, retrograde targeting of BDNF, and the upregulation of neuronal activity [73,75]. Ser478 located in the juxtamembrane region of TrkB is phosphorylated by cell division protein kinase 5 (CDK5), which activates Rac 1 that is involved in the BDNF-dependent remodeling of dendritic spines [75]. Signaling cascades emanating from TrkB phosphorylation lead to the synaptic remodeling of GABAergic neurons after nicotine exposure, which may act to downregulate or upregulate nicotine-induced behaviors depending on the conditions of nicotine exposure [78].

Unlike mature-BDNF, pro-BDNF facilitates interactions between p75NTR and several adaptor proteins, such as neurotrophin receptor-interacting factor (NRIF), melanoma-associated antigen (MAGE), neurotrophin receptor p75 interacting MAGE homologue (NRAGE), Schwann cell factor 1 (SC1), and RhoGDI [79], which function by activating the expression of pro-survival or pro-apoptotic genes [80]. However, it is unclear whether p75NTR-linked signaling cascades contribute to behavioral sensitization or nicotine-seeking behavior.

## 7. BDNF Differently Regulates Behavior According to the Conditions of Nicotine Exposure

Nicotine increases the anterograde and retrograde targeting of BDNF in striatal glutamate terminals and GABAergic neurons, respectively, and anterograde targeting of BDNF occurs in cortical neurons projecting to the striatum [81,82,83,84]. Evidence shows nicotine exposure induces the over-expression of *BDNF mRNA* in cultured cortical neurons [84]. For example, chronic treatment with the nicotine analog, choline, increased the expression and release of BDNF from cultured cortical neurons [83]. Stimulation of α7 nAChRs in the SH-SY5Y cell line increased BDNF synthesis and secretion [82,83], whereas treatment with MLA, an α7 nAChR antagonist, reduced the nicotine-induced BDNF over-expression in this cell line [28]. These findings suggest that nicotine-induced α7 nAChR stimulation is responsible for the expression of BDNF, which facilitates anterograde BDNF release to the striatum from cortices. A few studies have examined the effects of nicotine on retrograde BDNF release from corticostriatal nerve terminals [85]. Treatment with glutamate increases BDNF release in dendrites of cultured hippocampal neurons [86,87].

The role played by BDNF in the regulation of nicotine-induced behaviors is not unequivocal. Growing evidence demonstrates BDNF in the striatum regulates nicotine-induced behaviors in ways that depend on the nature of nicotine exposure. Intra-striatal infusion of BDNF in rats augments increases in locomotor activity induced by repeated cocaine exposure [88,89]. Intra-NAc infusion of a TrkB antagonist reduced increases in cocaine-seeking caused by repeated exposure [90,91]. Systemic administration of a TrkB antagonist prior to repeated exposure to nicotine reduced nicotine intake, motivation, and reinstatement of nicotine-seeking [90]. In addition, infusion of exogenous BDNF after repeated exposure to nicotine increased new synapse formation in dendritic spines of the NAc and CPu [92], which suggests BDNF elevation after repeated exposure to nicotine increases synaptic strength in the striatum, and thus, enhances behavioral sensitization and nicotine-seeking behavior.

In contrast, intra-NAc infusion of BDNF during nicotine withdrawal decreased nicotine-induced increases in the reinstatement of self-administration in rats [93]. Intra-CPu infusion of BDNF prior to nicotine challenge reduced nicotine-induced increases in locomotor activity and stereotypy movement [94]. As observed for nicotine withdrawal, intra-NAc infusion of a TrkB antagonist during cocaine withdrawal before cue-induced reinstatement increased cocaine self-administration in rats [91]. Furthermore, cue-induced increase in reinstatement evoked by cocaine administration was decreased by intra-NAc infusion of BDNF during cocaine withdrawal [91]. Intra-PFC infusion of BDNF during early withdrawal reduced cocaine-induced increases in self-administration in rats [91,95,96,97]. These findings suggest that GABAergic activity decreases during nicotine withdrawal. However, BDNF infusion into the NAc lowered GABAergic activity to the basal level by activating BDNF-mediated TrkB signaling cascades [91,96,97], which is believed to be why exogenous BDNF infusion in brain followed by nicotine challenge reduces behavioral sensitization and nicotine-seeking as compared with nicotine challenge alone. This notion is supported in part by the finding that exogenous BDNF infusion in the PFC during early cocaine withdrawal increases ERK phosphorylation [97].

The effects of BDNF on glutamatergic neurotransmission, bidirectional BDNF releases in the striatum, behavioral sensitization after repeated exposure to nicotine, and the effects of nicotine withdrawal are summarized in Figure 1. Repeated exposure to nicotine increases glutamate release in the striatum by stimulating α7 nAChRs, and dopamine may help to increase glutamate release [7,8,9]. In concert with increased glutamate release, BDNF release in the striatum is also enhanced by repeated exposure to nicotine [16,19,20]. Furthermore, this increase in BDNF release, resulting from anterograde and retrograde secretion, stimulates BDNF-mediated TrkB signaling cascades in GABAergic neurons, and thus, leading to psychomotor sensitization and nicotine-seeking behavior [21,69,74,91]. In contrast, BDNF infusion in the NAc returned hypo-activated GABAergic neurons during nicotine withdrawal to the basal level, which leads to downregulate nicotine challenge-induced increases in behaviors [96,97]. For this reason, infusion of BDNF during nicotine withdrawal might provide an effective means of controlling nicotine-induced relapse after withdrawal. Therefore, understanding of the molecular activities of BDNF and associated molecules in TrkB-mediated signaling cascades may aid the identification of potential therapeutic targets for the prevention of nicotine relapse.

## Figures and Tables

**Figure 1 ijms-20-02943-f001:**
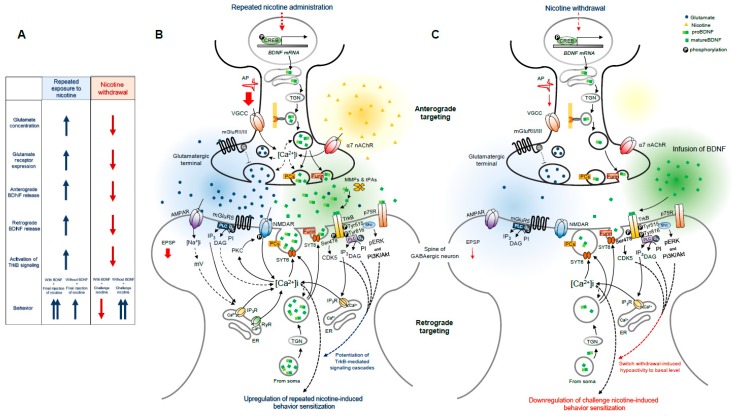
(**A**) Changes in glutamatergic neurotransmission, bidirectional BDNF release, BDNF-mediated signaling cascades, and behavioral sensitization after repeated exposure to nicotine and nicotine withdrawal. (**B**,**C**) Proposed cellular mechanisms underlying repeated nicotine exposure and nicotine withdrawal mediated effects in GABAergic neurons are schematically depicted and compared. Exogenous BDNF infusion into the striatum during nicotine withdrawal returned hypoactivated GABAergic neurons to the basal level by activating TrkB-linked signaling cascades [91,96,97]. Switched neural activity caused by BDNF does not alter after challenge nicotine administration. Putative interactions are discussed in the text. Solid and broken arrows represent direct and indirect stimulations of downstream molecules or behaviors, respectively. AP, action potential; [Ca^2+^], Ca^2+^ concentration; EPSP, excitatory postsynaptic potential; IP_3_R, inositol trisphosphate receptor; mV, membrane potential; [Na^+^], Na^+^ concentration; PKC, protein kinase C; RyR, ryanodine receptor; SYT, synpatotagamin; tPAs, tissue-plasminogen activators; VGCC, voltage-gated calcium channel.

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
