# Peer review of "Interactions of Glutamatergic Neurotransmission and Brain-Derived Neurotrophic Factor in the Regulation of Behaviors after Nicotine Administration"

_ijms, 2019, doi:10.3390/ijms20122943_

Reviewer 1 Report

Please change ....

N-AchRs to nAchRs(line 67)

M-GluRs to mGluRs (line 110)

Author Response

We uploaded the file entitled 'ChoeResRev1'.

Reviewer 2 Report

List of abbreviations is missing, therefore a lot of the abbreviated words are not clearly understandable.

The words labelled in yellow are suggested for insertion by the reviewer.

Line 16: …when alpha7 nicotinic…

Line 35: instead „stimulations”, use „elevated transmitter levels

Line 37: …to other parts of basal ganglia. Striatum is one part of the basal ganglia: the outputs of striatum are running  to pallidum (medial and lateral) and to the substantia nigra.

Line 38. Nicotine exposure (delete „interestingly”)…

Line 40: evidence demonstrates that BDNF…

Clarify that glutamate terminals in the striatum are the afferents originating in the neocortex and thalamus. Also clarify, that dopaminergic inputs of the CPu are more complicated than stated here: they originate from SN pars compacta, pars reticulata and VTA (see Paxinos: The Rat Nervous System, latest edition).

Line 47-52: dorsal and ventral striatum are defined mainly in human – not in rat (Blumenfeldt, 2010). In human, the ventral striatum involves the head of the caudate nucleus AND the nucleus accumbens. In rat, the nucleus accumbens is listed as part of the striatum (Gerfen, 2004).

Line 73-74: „For instance, blockade…..” this sentence has no sense.

Lines 79-80:  „It has also been reported… glutamate release in the CPu innervating neurons from the PFC.” this has no sense – tha authors wanted to say that the corticostriatal axon terminals release more glutamate?

Line 81: what is „indirect glutamate release”?

Lines 84-92: please define „behavioral sensitization” and „psychomotor behaviors”

Instead „nicotine-seeking behavior” use nicotine addiction (everywhere in the text)

Lines 93-94: „nicotine-induced behavior” needs clarification

Line 96: …Two types (or families) of glutamate receptors are expressed in the striatum: the iGluRs and the mGluRs.

line 99: …is known to upregulate…

Line 100: …demonstarte that increased…

Line 101: Delete „For instance” Repeated nicotine…

Line 103: …reveals that local nicotine…

Line 104: … suggest that stimulation

Line 106: …findings that nicotine addiction behavior…

Line 110: The mGluRs…

Line 114: In turn, IP3 increases…

Line 122: instead „glutamate overflow” use extracellular glutamate

Lines 125-130: two sentences have no sense in this form. „It has been shown……feedback control manner.”

Line 145: Binding of the htt complex to vesicles… (omit „formed”)

Secretion vesicles ? Secretory granule containing vesicles? What are the differences? Short definition needed.

Lines 149-153: these sentences should be: „Subsequent exocytosis triggered by the stimulation of α7 nAchRs by nicotine binding results in the relase of a mixture of pro- and mature BDNF []. Released pro-BDNF…..which are located outside of neurons []. These extracellular enzymes….in striatal GABAergic neurons.”

Line 155: Apart from anterograde BDNF release…

Lines 156-159: ..axonal transport. The stimulation of glutamate receptors by nicotine-induced glutamate release facilitates BDNF secretion into synaptic clefts…   Although the cellular mechanisms responsible for the retrograde release of BDNF from the dendritic spines in the striatum…

Lines 163-164: I do not know SANRE proteins, but I do know SNARE proteins which are syntaxin and synaptobrevin. The authors should consider this problem.

Lines 170-173: Suggested: …Furthermore, BDNF infusion into the NAc stimulates…. results in the phosphorylation of tyrosine and serine of TrkB at positions Tyr515, Tyr816 and Ser478 [].

Lines 174-175: „Furthermore, ..The activation of Pi3K/Akt and Ras signalling pathways leads to ERK phophorylation….

Line 178: …increases of Ca2+ release…

Line 184: …may act either to downregulate….

Line 191: …nicotine addiction behavior.

Line 192: ….regulates behavior according….

Choline is listed as an agonist of nicotine, because it is acting on nicotinic ACh receptors. Analogs (analogues) in addiction literature are those chemicals which can mimic the action of nicotine (e.g. cytisine).

Line 200: …These findings suggest that nicotine-induced…

Line 203: …retrograde BDNF release from corticostriatal axons.

Lines 203-204: „In GABAergic neurons…” this sentence has no sense for me.

Line 206: …straightforward – instead of this word I should use „completely clear” or „unequivocal”

Line 212: …reinstatement… instead of this word I should use „recommence” or „started all over again”

Lines 212-214: In addition, infusion… this sentence is not clear. The dendritic spines became larger? New synapses were formed? The two phenomena are related?

Line 219:  …stereotypy movement: what is this?

Line 225: These findings suggest that GABAergic activity was/is decreased during nicotine withdrawal. The next sentence („However, BDNF administered…”) is not clear: what is „basal level” activity? This complete paragraph (lines 217-231) is a bit confusing, because nicotine and cocaine are mentioned together, and it is not clear that the authors conclude from this or from the other.

Line 240: what is „hypo-activation”?

Figure 1. A: what is „challenge nicotine”? The reader needs a short explanation (probably this is a single injection in the „withdrawal” animals – but when, what dose, intravenous or intracerebral?)

B, C: what is „nicotine-induced behavior sensitization”? The reader needs a short explanation of this.

Figure caption: the authors should list the references on the basis of which this drawing was constructed. The Figure is informative, and good.

Author Response

We uploaded the file entitled 'ChoeResRev2'.

Round  2

Reviewer 2 Report

Thank you for the answers. I accept your points completely. The manuscripts has been improved considerably.